# Optimization of Thermoplastic Blend Matrix HDPE/PLA with Different Types and Levels of Coupling Agents

**DOI:** 10.3390/ma11122527

**Published:** 2018-12-12

**Authors:** Alessia Quitadamo, Valérie Massardier, Carlo Santulli, Marco Valente

**Affiliations:** 1Department of Chemical and Material Engineering, Università di Roma La Sapienza, via Eudossiana 18, 00184 Rome, Italy; alessia.quitadamo@uniroma1.it; 2Ingénierie des Matériaux Polymères, INSA de Lyon, Université de Lyon 69003, 69621 Villeurbanne, France; valerie.massardier@insa-lyon.fr; 3School of Architecture and Design, Università di Camerino, Viale della Rimembranza, 63100 Ascoli Piceno, Italy; carlo.santulli@unicam.it

**Keywords:** blend, bio-derived polymers, compatibilization, thermoplastic matrix for composite

## Abstract

High-density polyethylene (HDPE) and poly(lactic) acid (PLA) blends with different ratios of both polymers, namely, 30:70, 50:50, and 70:30, were produced. Polyethylene-grafted maleic anhydride and a random copolymer of ethylene and glycidyl methacrylate were also considered as compatibilizers to modify HDPE/PLA optimal blends and were added in the amounts of 1, 3, and 5 wt.%. Different properties of the blends were evaluated by performing tensile tests and scanning electron microscopy to analyze blend and interfaces morphology. Moreover, thermomechanical analysis through differential scanning calorimetry, thermo-gravimetric analysis, and infrared spectroscopy were also performed. The blend containing equal amounts of HDPE and PLA seemed to present a good balance between amount of bio-derived charge and acceptable mechanical properties. This suggests that these blends have a good potential for the production of composites with lingo-cellulosic fillers.

## 1. Introduction

Blending is one of the simplest and widespread methods to improve polymer properties. In fact, low-cost processing, typical of common blend manufacturing, allows obtaining the desired properties and a high variety of products [1]. As a consequence, nowadays polymer blends represent around half of the total plastic production [2]. The main limit of a polymer blend is the mutual miscibility of polymers. In fact, as established by the second law of thermodynamics, the variation of free energy ΔG is usually positive because of the high polymerization degree of polymers (affecting the variation of entropy ΔS) and the poor affinity between polymers (affecting the variation of enthalpy ΔH) [3].

One of the main parameters affecting polymers miscibility is interfacial tension. In particular, higher interfacial tension leads to higher phase separation [4]. In spite of this limit, a polymer blend can be compatibilized in order to increase dispersion and adhesion between polymers [5]. Many strategies have been developed in this context, among which, the introduction of a compatibilizer is one of the most adopted [6]. Recently, higher sensitivity to problems of oil-based polymer pollution has developed, stimulating research on bio-derived polymer production for application as potential substitutes of oil-based polymers [7,8,9,10]. Among bio-derived polymers, poly(lactic) acid (PLA) seems to be one of the most studied and applied, thanks to its properties which are comparable to or in some cases higher than traditional polyolefins [11,12,13,14]. PLA, in fact, has been frequently selected as a bio-derived polymer to obtain blends with a high amount of biodegradable polymer, reducing the amount of polyolefin [15,16,17]. Moreover, depending on the compatibilizer added, an oil-based, bio-derived blend can be optimized and has specific properties. For example, a recent study compatibilized PLA/ high-density polyethylene (HDPE) polymer blends with cobalt stearate in view of a possible oxo-degradation process. This demonstrates the wide range of opportunities deriving from the addition of the right type and amount of compatibilizer [18]. Several compatibilizing methods have been studied in order to improve oil-based–bio-derived polymeric blends, such as the use of functional molecules for reactive compatibilization during extrusion or the addition of a commercial modified polymer as a coupling agent [19,20,21]. Usually, chemical compatibilization, obtained with the use of grafted polymers or random copolymers, is performed to reduce the size of the dispersed phase thanks to functional groups’ reactivity. In this way, a reduction of interfacial tension and of the coalescence impediment of the dispersed phase is possible [22].

The development of an oil-based, bio-derived thermoplastic blend was therefore the preliminary goal of this work. In particular, an optimized high-density polyethylene and poly(lactic) acid blend could be produced in order to obtain oil-based–bio-derived thermoplastic blends with high amounts of bio-derived charge, while maintaining good mechanical properties.

Two kinds of compatibilizers were tested in order to improve the blend properties. Polybond 3029 and Lotader AX8840 seemed to be effective, thanks to the presence of maleic anhydride grafted on polyethylene chains for the former and polyethylene random copolymer with glycidyl methacrylate for the latter.

## 2. Materials and Methods

Eraclene MP90, commercial name of high-density polyethylene (HDPE) from ENI (Versalis, San Donato Milanese, Italy), was selected as the oil-based polymer. Among its properties are: a melt flow index (MFI) of 7 g/10 min (190 °C/2.16 kg), a nominal mass of 0.96 g/cm³, a tensile strength of 21 MPa, a tensile modulus of 1.2 GPa, and a Shore D hardness of 50. Poly(lactic acid) (PLA) Ingeo Biopolymer 3251D from Nature Works (Minnetonka, MN, USA) was selected as a bio-derived thermoplastic polymer, with an MFI of 35 g/10 min (190 °C/2.16 kg). This polymer is characterized by density 1.24, crystalline melting temperature in the range 155–170 °C, and a glass transition temperature in the range 55–60 °C. Polybond 3029 (Addivant, CT, USA) was selected as an additive, suitable for cellulosic fillers. In fact, Polybond 3029 is a maleated polyethylene with a melt flow index of 4 g/10 min (190 °C/2.16 kg), and a maleic anhydride (MA) content of 1.7 wt.% (high). Generally, it is sold as pellets of 3–4 mm diameter. Lotader AX8840 was selected with the same purpose. It is a random copolymer of ethylene and glycidyl-methacrylate (GMA), with a melt flow index of 5 g/10 min (190 °C/2.16 kg). The GMA content is about 8 wt.%. PLA was dried for one night at 80 °C in order to avoid possible bubble formation due to water evaporation during the production process.

A Micro 15 Twin-screw DSM research extruder (Xplore Instruments BV, 6160 MD Geleen, The Netherlands) was used in order to produce the samples. Temperature of 180 °C, screw speed of 75 rpm, nitrogen atmosphere, and a resident time of 4 min in the extruder were selected to avoid PLA degradation during the process [11,12,13,14,15,16,17,18,19,20,21,22,23,24].

Injection moulding was used to obtain dog-bone specimens, with a mould temperature of 55 °C and pressure parameters depending on polymer viscosity. For each family of samples, 10 specimens were produced. Table 1 sums up the formulations produced.

### 2.1. Tensile Tests

Tensile tests were performed in accordance with the ASTM D638-14 standard, using Zwick/Roell Z010 (ZwickRoell GmbH & Co. KG, D-89079 Ulm, Germany), with a load cell of 10 kN and a 50 N preload. The crosshead speed was 5 mm/min. The tensile tests were performed on five dog-bone samples per series, with a gauge length section of 30 × 4 × 2 mm^3^ (L × W × T). For each family, five samples were tested.

### 2.2. Scanning Electron Microscopy (SEM)

The samples were observed with a Hitachi S2500 25 kV scanning electron microscope (Hitachi, Krefeld, Germany) in order to analyze blend morphology and interfaces. The samples were sputter-coated with gold particles before surface characterization.

### 2.3. Quartering

The samples produced, in the majority of cases, were characterized by some heterogeneity because of multiphase matrices. In order to obtain reliable results from the thermal analysis and analyze a representative number of samples, a cryogenic mill was adopted to obtain samples in the form of powders. A subsequent statistical approach, quartering, was used to select an exemplary number of samples used for chemical and thermal analysis. This method was based on the separation of the total amount of charge in four parts equal in weight. Then, two parts at the opposite side were mixed together, and the other two were separated.

### 2.4. Differential Scanning Calorimetry (DSC)

Differential Scanning Calorimetry (DSC) tests were performed on a Q20 Thermal Analysis instrument (TA Instruments, New Castle, DE, USA) from 25 °C to 180 °C at 10 °C/min under a nitrogen flow of 50 mL/min^−1^. Two cycles were performed with a 4 min interval between them at 180 °C to eliminate trace of thermal history. The first cycle provided information about properties after injection moulding, while the second one gave material’s properties. Cold crystallization, melting, crystallization parameters (temperature and enthalpy), and glass transition temperatures were analyzed.

### 2.5. Thermogravimetric Analysis (TGA)

Thermogravimetric Analysis (TGA) tests were carried out on a Q500 Thermal Analysis instrument (TA Instruments, New Castle, DE, USA) up to 600 °C, with a scanning temperature of 10 °C/min under a nitrogen flow of 50 mL/min^−1^. From this analysis, we derived temperatures at which degradation started (T_onset_), evaluated through the extrapolated onset temperature from the TGA curve and the ∆m, i.e., mass variation, during the test.

### 2.6. Attenuated Total Reflection–Fourier Transform Infrared (ATR–FTIR) Analysis

Attenuated Total Reflection–Fourier Transform Infrared (ATR-FTIR) tests were carried out to evaluate the interactions between HDPE, PLA, and the compatibilizers. The tests were performed with a thermo-scientific Nicolet IS10 spectrometer (Thermo Fisher Scientific, Waltham, MA, USA) with a spectral range 4000–400 cm^–1^ and 32 scans.

## 3. Results

### 3.1. Tensile Tests

HDPE/PLA blends’ properties were first analyzed through tensile tests of HDPE70/HDPE30, HDPE50/PLA50, and HDPE30/PLA70 to evaluate mechanical performance and identify the influence of polymer blend. Table 2 sums up the results of these three blends compared to neat HDPE and PLA, while Figure 1 displays the tensile tests curves of the HDPE/PLA blends.

The presence of 30 and 50 wt.% PLA allowed reaching a good tensile stiffness and strength with respect to neat HDPE, while at the same time keeping acceptable elongation at break. This is a significant result considering the considerable amount of brittle polymer (PLA) blended with HDPE. In contrast, increasing the amount of PLA up to 70 wt.% strongly reduced the elongation at break, at a level even lower than that for neat PLA, while offering a tensile strength and stiffness near to those of neat PLA [25]. Starting from these results, HDPE50/PLA50 was selected as a promising blend, offering a good compromise of mechanical properties and amount of bio-based material. As a consequence, the analysis of blends with 50 wt.% PLA were considered for deeper studies. The use of three different percentages of compatibilizers was investigated to elucidate their effects on HDPE50/PLA50 and to identify the most suitable compatibilizer and its amount. In particular, Lotader AX8840 and Polybond 3029 were investigated as compatibilizing agents in the amounts of 1, 3, and 5 wt.%. Table 3 and Figure 2 display the main tensile tests results.

Lotader AX8840 and Polybond 3029 were selected because of their ability to interact with both polyethylene and polymers with polar groups, such as poly(lactic) acid. The addition of Lotader AX8840 seemed to be very effective in improving the compatibility between HDPE and PLA, compared to other compatibilizing agents [26,27]. Significant improvement in elongation at break was displayed for all compatibilizer percentages, despite the presence of a high percentage of brittle polymer like PLA. At the same time, slight reductions in elastic modulus and tensile strength were displayed for the highest compatibilizer percentage. Moreover, Polybond 3029 similarly displayed a reduction of σ and ε when 5 wt.% was added. As a consequence, an optimal amount of Lotader AX8840 and Polybond 3029 need to be applied. In both cases, the best results were obtained for 3 wt.% of compatibilizer.

### 3.2. Scanning Electron Microscopy (SEM)

Scanning electron microscopy (SEM, Hitachi, Krefeld, Germany) analyses aimed at investigating both the compatibility between HDPE and PLA and the influence of compatibilizers in the blend. HDPE/PLA SEM images displayed the typical immiscible blend morphology obtained when mixing hydrophobic (polyethylene) and hydrophilic (poly (lactic) acid) compounds (Figure 3). In fact, a visible phase separation between HDPE and PLA was displayed, as expected for polymers with different hydrophilicity, a weak interface between HDPE and PLA was evident. The problem of compatibilization between polyethylene and poly(lactic) acid polymers is of paramount importance in recent resin manufacturing and has been partially addressed with interfacially localized catalysts, based for example on stannous octoate [28]. However, the problem appear in general terms far from being totally resolved.

The addition of compatibilizers to the HDPE50/PLA50 blend allowed a reduction in dimension of the different phases, suggesting a higher compatibility between HDPE and PLA. This result was more evident for Lotader AX8840 (Figure 4a) than for Polybond 3029 (Figure 4b). In fact, a higher phase separation and easier distinction between polymers were displayed by samples compatibilized with Polybond 3029. The higher affinity of PLA for Lotader AX8840 was already shown in our previous work [11], revealing the presence of a smaller spherical secondary phase for Lotader AX8840 than for Polybond 3029 when blended with PLA. The reactive groups of the compatibilizers can interact with the hydroxyl and/or carboxyl groups of PLA, while the ethylene chain of the compatibilizers can easily be mixed with HDPE. These results are in agreement with the FTIR analyses.

### 3.3. Differential Scanning Calorimetry (DSC)

Differential Scanning Calorimetry was useful for analyzing the effects of blending HDPE with PLA and those of adding compatibilizers. ΔHm_PE_ (J/gPE) and ΔHm_PLA_ (J/gPLA) refer the enthalpy values to the exact amount of polyethylene and poly(lactic) acid in the samples. For example, HDPE50/PLA50 displayed ΔHm_PE_ of 113 J/g, but dividing by 0.5 is needed to obtain the real amount of enthalpy for HDPE. As a consequence, the effective ΔHm_PE_ was 226 J/g_PE_. Table 4 sums up the DSC results for HDPE, PLA, and HDPE50/PLA50.

The typical PLA cold crystallization process [29] was still evident in HDPE50/PLA50 blend. Moreover, a slightly higher crystallinity of the HDPE phase was demonstrated, in agreement with both higher mechanical properties of the blend and the occurrence of phase separation because of the presence of crystals [4]. The effects of Lotader AX8840 and Polybond 3029 on HDPE50/PLA50 properties were also analyzed (Table 5). Both Polybond 3029 and Lotader AX8840 addition revealed the presence of cold crystallization. Higher enthalpies values were measured for Lotader AX8840 addition than for Polybond 3029, but in both cases, the influence of adding different amounts of compatibilizers was not evident. A possible interpretation for higher enthalpies is that the addition of compatibilizers results in an increased PLA chain mobility with respect to neat HDPE50/PLA50.

A more evident effect of compatibilizer addition was displayed by Lotader AX8840, with a lower melting enthalpy for the HDPE phase. This result suggests an interaction between HDPE and PLA through Lotader AX8840, hindering HDPE macromolecules mobility [30].

### 3.4. Thermogravimetric Analysis (TGA)

In order to evaluate the thermal stability of the polymers, an analysis was performed to measure the degradation onset temperature (T_onset_) (Table 6). PLA revealed a lower thermal stability compared to HDPE. In fact, T_onset_ was around 319 °C for PLA and 458 °C for HDPE, which is in agreement with literature results [31]. Blending HDPE and PLA (HDPE50/PLA50) resulted in a T_onset_ near to that of neat PLA, confirming a reduced thermal stability of the blend (322 °C) with respect to neat HDPE (458 °C). All formulations displayed a complete degradation of polymers without the formation of a residual char (100% of mass variation, ∆m, between the mass of the sample before the test and the residual mass after the test). The addition of Lotader AX8840 increased blend thermal stability, with higher T_onset_ when increasing Lotader AX8840 amount. The blend formulations, either compatibilized or not, displayed two separate thermal degradations, the first referring to poly(lactic) acid, and the second to polyethylene. A maximum rate of weight loss corresponded to each degradation. The blend formulations exhibited two separate temperatures of the peak value for the first derivative of the TGA curve (T_DTG_).

### 3.5. Attenuated Total Reflection–Fourier Transform Infrared Spectroscopy (ATR–FTIR)

Infrared spectroscopy (Thermo Fisher Scientific, Waltham, MA, USA) was used to evaluate the interactions between polymers, whose main results are displayed in Figure 5. As expected, HDPE50PLA50 did not display peak variations, confirming the presence of an immiscible blend without interactions between HDPE and PLA. The addition of Lotader AX8840 slightly shifted the typical ester peak of PLA (1749 to 1752 cm-1), suggesting interactions between glycidyl methacrylate and the C=O group of PLA. Polybond 3029 addition, on the contrary, did not display variation of HDPE50/PLA50 peaks, confirming the hypothesis of poor interactions between maleic anhydride and the C=O group of PLA.

Further research on compatibilization could involve the use of nanostructures, such as SiO_2_ nanoparticles, graphene platelets, or carbon nanotubes [32].

## 4. Conclusions

After a preliminary study on HDPE/PLA blends, the blend containing equal amounts of HDPE and PLA appeared to be the most suitable towards keeping good mechanical properties and a significant reduction of non-bio-derived charge. The addition of a compatibilizer, especially Lotader AX8840, with its high content of glycidyl methacrylate, seemed to increase the homogeneity of the blend. An appropriate percentage of a compatibilizer has to be selected in order to optimize a blend’s properties. In fact, 3 wt.% of a compatibilizer seemed to optimize the mechanical properties (strength and plasticity) and the affinity between HDPE and PLA. SEM images revealed a typical immiscible morphology for HDPE and PLA when blended without a compatibilizer. Both Lotader AX8840 and, more moderately, Polybond 3029 seemed to increase the homogeneity of the blend thanks to the interaction of functional groups with PLA, which is in agreement with the FTIR results. Further analyses have to be done in order to evaluate the biodegradation behavior of oil-based–bio-derived polymer blends at the ratios analyzed in this study.

## Figures and Tables

**Figure 1 materials-11-02527-f001:**
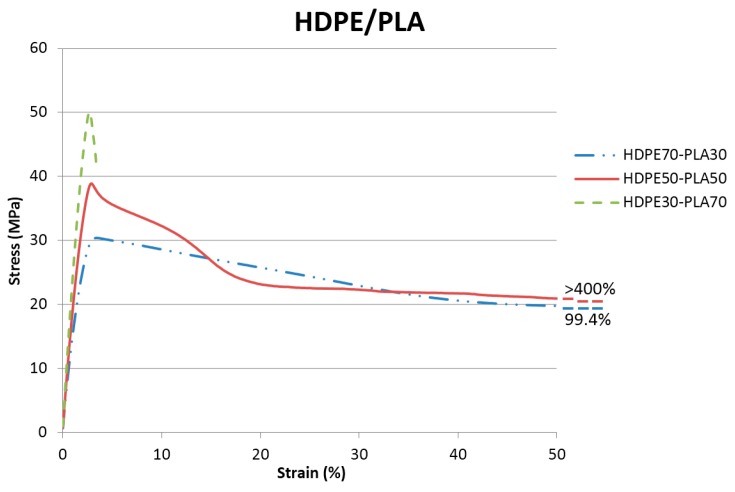
Tensile tests graphs of HDPE70/PLA30, HDPE50/PLA50, and HDPE30/PLA70.

**Figure 2 materials-11-02527-f002:**
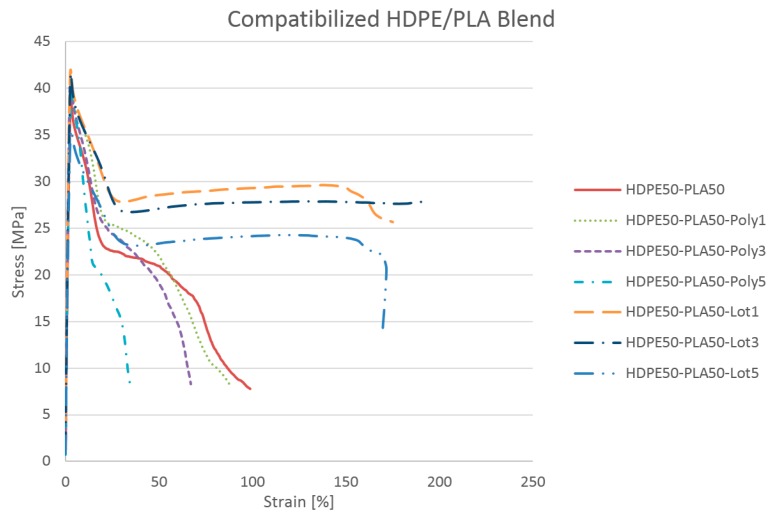
Effect of Lotader AX8840 and Polybond 3029 at 1, 3, and 5 wt.% on the PE50/PLA50 blend.

**Figure 3 materials-11-02527-f003:**
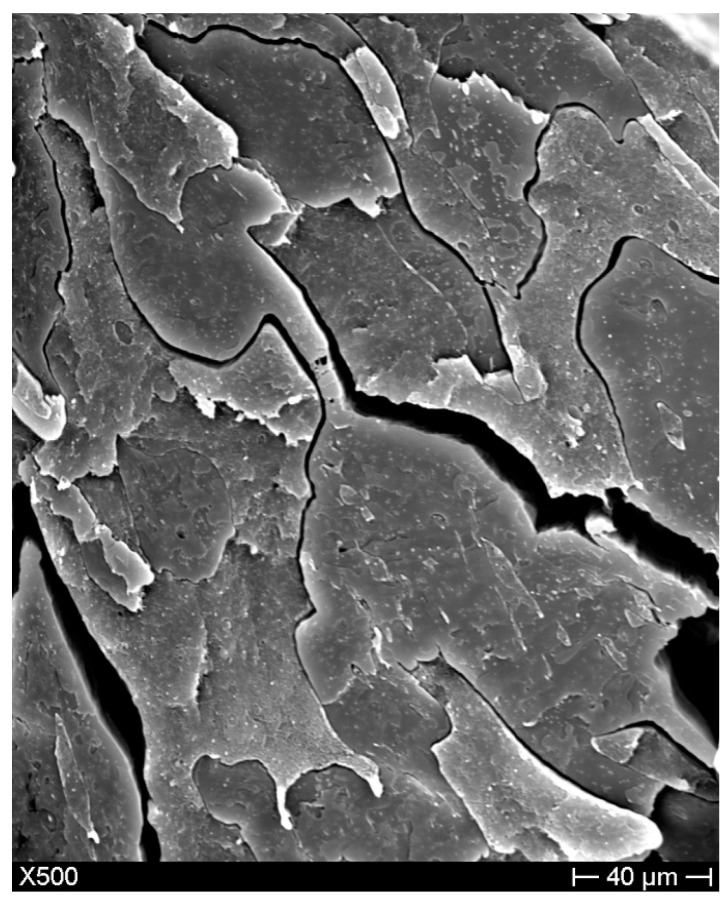
Morphology of the cryo-fractured surface of the HDPE50/PLA50 blend.

**Figure 4 materials-11-02527-f004:**
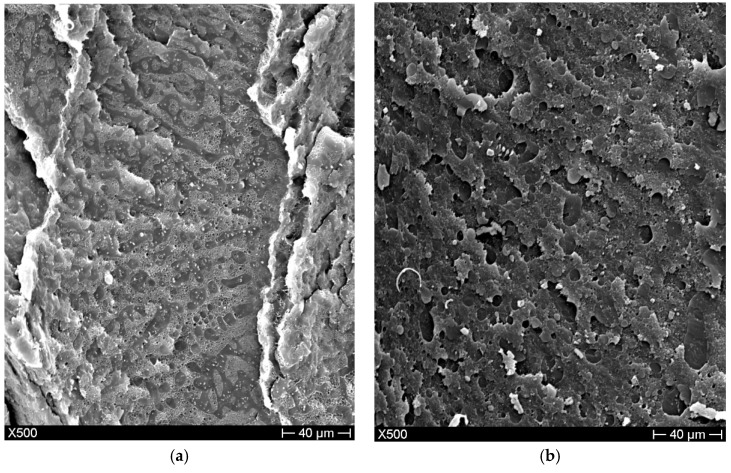
Blend morphology of the cryo-fractured surface of (**a**) HDPE50/PLA50-Lot3, (**b**) HDPE50/PLA50-Poly3.

**Figure 5 materials-11-02527-f005:**
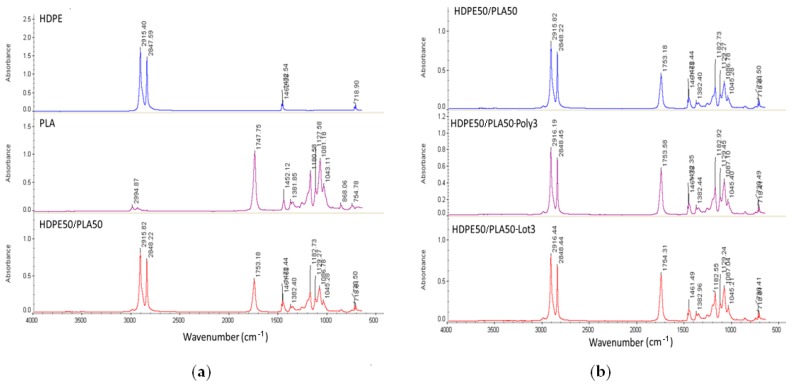
Attenuated total reflection (ATR)–FTIR spectra of: (**a**) HDPE, PLA, and HDPE50/PLA50, (**b**) HDPE50/PLA50, HDPE50/PLA50-Lot3, and HDPE50/PLA50-Poly3.

**Table 1 materials-11-02527-t001:** Formulations produced (10 samples for each family). HDPE: high-density polyethylene, PLA: poly(lactic) acid.

Samples	HDPE (%)	PLA (%)	Polybond 3029 (%)	Lotader AX8840 (%)
HDPE70/PLA30	70	30		
HDPE50/PLA50	50	50		
HDPE30/PLA70	30	70		
HDPE50/PLA50-Poly1	49.5	49.5	1	
HDPE50/PLA50-Poly3	48.5	48.5	3	
HDPE50/PLA50-Poly5	47.5	47.5	5	
HDPE50/PLA50-Lot1	49.5	49.5		1
HDPE50/PLA50-Lot3	48.5	48.5		3
HDPE50/PLA50-Lot5	47.5	47.5		5

**Table 2 materials-11-02527-t002:** Tensile tests results of HDPE/PLA blends without compatibilizers.

Samples	E (GPa)	σ (MPa)	ε (%)
HDPE	1.16 ± 0.08	21.59 ± 0.18	>400
PLA	3.04 ± 0.02	57.34 ± 1.00	7.1 ± 0.3
HDPE70/PLA30	1.51 ± 0.05	30.76 ± 0.73	>400
HDPE50/PLA50	1.88 ± 0.05	38.73 ± 0.18	99.4 ± 2.1
HDPE30/PLA70	2.41 ± 0.05	49.51 ± 0.60	2.3 ± 0.5

**Table 3 materials-11-02527-t003:** Tensile tests results for HDPEPLA blends with Lotader AX8840 and Polybond 3029 in different amounts (1, 3, 5 wt.%). The purple colour indicates or Polybond 3029 addition, while the green colour is for Lotader AX8840.

Samples	E (GPa)	σ (MPa)	ε (%)
HDPE50/PLA50	1.88 ± 0.05	38.73 ± 0.18	99.4 ± 2.1
HDPE50/PLA50-Poly1	2.24 ± 0.76	43.30 ± 2.76	86.2 ± 17.3
HDPE50/PLA50-Poly3	2.31 ± 0.14	42.80 ± 2.65	71.9 ± 46.9
HDPE50/PLA50-Poly5	1.92 ± 0.03	39.74 ± 0.41	34.1 ± 13.1
HDPE50/PLA50-Lot1	2.18 ± 0.21	40.70 ± 3.99	175.3 ± 84.2
HDPE50/PLA50-Lot3	2.14 ± 0.07	41.70 ± 1.93	193.0 ± 59.1
HDPE50/PLA50-Lot5	1.75 ± 0.13	34.28 ± 1.52	173.2 ± 56.2

**Table 4 materials-11-02527-t004:** Differential scanning calorimetry (DSC) results for HDPE, PLA, and HDPE50/PLA50.

	ΔHcc_PLA_	Tcc	ΔHm_PE_	Tm_PE_	ΔHm_PLA_	Tm_PLA_	Tg_PLA_
	(J/g_PLA_)	(°C)	(J/g_PE_)	(°C)	(J/g_PLA_)	(°C)	(°C)
HDPE	-	-	215	134	-	-	-
PLA	7	98	-	-	41	168	61
HDPE50/PLA50	8	97	226	132	40	168	62

**Table 5 materials-11-02527-t005:** DSC results of HDPE50/PLA50 with 1, 3, 5 wt.% of Lotader AX8840 and Polybond 3029.

	ΔHcc_PLA_	Tcc	ΔHm_PE_	Tm_PE_	ΔHm_PLA_	Tm_PLA_	Tg_PLA_
	(J/g_PLA_)	(°C)	(J/g_PE_)	(°C)	(J/g_PLA_)	(°C)	(°C)
HDPE50/PLA50	8	97	236	132	40	168	62
HDPE50/PLA50-Poly1	15	94	195	132	45	168	61
HDPE50/PLA50-Poly3	15	100	190	132	41	168	61
HDPE50/PLA50-Poly5	16	101	198	132	41	168	61
HDPE50/PLA50-Lot1	20	103	192	132	40	168	61
HDPE50/PLA50-Lot3	21	103	182	132	41	168	61
HDPE50/PLA50-Lot5	20	104	192	132	36	168	61

**Table 6 materials-11-02527-t006:** TGA results for HDPE/PLA blends with different compatibilizer percentages and HDPE50/PLA50 matrix composites with 3 wt.% of compatibilizer. T_onset_ (°C) was evaluated with the extrapolated onset temperature from the TGA curve. T_DTG_ (°C), temperature of maximum differential thermogravimetric analysis (DTG) curve peaks; ∆m (%) is the mass variation percentage between sample’s total mass before the test and the residual mass after the test.

	T_onset_ (°C)	T_DTG_ (°C)	∆m (%)
HDPE	458	474	100
PLA	319	351	100
Poly	459	480	100
Lot	434	464	100
HDPE50/PLA50	322	345/471	100
HDPE50/PLA50-Poly1	316	351/444	100
HDPE50/PLA50-Poly3	322	358/438	100
HDPE50/PLA50-Poly5	313	342/432	100
HDPE50/PLA50-Lot1	325	350/470	100
HDPE50/PLA50-Lot3	325	350/468	100
HDPE50/PLA50-Lot5	334	351/474	100

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
