# Peer review of "Optimization of Thermoplastic Blend Matrix HDPE/PLA with Different Types and Levels of Coupling Agents"

_materials, 2018, doi:10.3390/ma11122527_

Round 1
Reviewer 1 Report
The work carried out by Quitadamo et al. is investigating an important topic. Although the overall quality of the manuscript is enough to be accepted for publication in Materials, the following comments need to be considered:
1. The selection of conditions used for the blend fabrication, temperature and screw speed, should be supported by a suited reference.
2. The mechanism that responsible for the compatibilization in this blend should be introduced.
3. The figures should be reproduced in a more clear way. For example, the text used on the axes in Fig. 5, are not readable.
4. For the application of this composition, it would be useful also to investigate the impact properties.
5. One English spelling should be used, American or British, for example mould and behavior.
6. English is readable but it should be more professional.
Author Response
Open Review 1
( ) I would not like to sign my review report
(x) I would like to sign my review report
English language and style
( ) Extensive editing of English language and style required
( ) Moderate English changes required
(x) English language and style are fine/minor spell check required
( ) I don't feel qualified to judge about the English language and style
Comments and Suggestions for Authors
The work carried out by Quitadamo et al. is investigating an important topic. Although the overall quality of the manuscript is enough to be accepted for publication in Materials, the following comments need to be considered:
1. The selection of conditions used for the blend fabrication, temperature and screw speed, should be supported by a suited reference.
Three references (11-19-20) have been added to justify processing parameters.
2. The mechanism that responsible for the compatibilization in this blend should be introduced.
Chemical blending has been added in the introduction “Usually, chemical compatibilization, obtained with the use of grafted polymers or random copolymers, is performed to reduce the size of dispersed phase thanks to functional groups reactivity. In this way, a reduction of interfacial tension and the coalescence impediment of the dispersed phase is possible [22].”
SEM “Reactive groups of compatibilizers can interact with hydroxyl and/or carboxyl groups of PLA, while ethylene chain of compatibilizers can easily be mixed with HDPE. These results are in agreement with FTIR analyses.” Explanation of possible mechanisms responsible for higher compatibilization
3. The figures should be reproduced in a more clear way. For example, the text used on the axes in Fig. 5, are not readable.
Figure 5 has been modified.
4. For the application of this composition, it would be useful also to investigate the impact properties.
Impact properties were not evaluated because with our extrusion and injection moulding equipment we cannot produce samples for this test. However, future studies should concern these analyses because of the importance of this investigation for these materials.
5. One English spelling should be used, American or British, for example mould and behavior.
English spelling has been used.
6. English is readable but it should be more professional.
English has been reviewed.
Reviewer 2 Report
The manuscript reported by the present authors investigated the preparation and characterization of PLA/HDPE in the presence of different coupling agents. The manuscript is well-done and arranged. The authors have done enough experiments to show the study is relevant and adequate for publications. Some points and clarifications, however, should be added and discussed to make this manuscript suitable for publication in Materials.
1. The mechanical results of PLA/PE blends with/without coupling agents should be discussed and compared with previous investigations in literature. The lack of discussion makes difficult to determine the scientific contribution of the present work.
2. It would be useful for the readers if the authors give more insights and explanations on the compatibilization mechanism.
3. Relevant literature should be cited. The following examples are highly recommended;
[1] E. Javadi, A. Babaei, M. Nouri, Correlation of the morphological and mechanical properties of a biodegradable blend based on polylactic acid, Journal of Macromolecular Science, Part B: Physics. 56 (2017) 194-201.
[2] K. Hamad, M. Kaseem, M. Ayyoob, J. Joo, F. Deri, Polylactic acid blends: The future of green, light and tough, Progress in polymer Science. 85 (2018) 83-127.
[3] Jesús Manuel Quiroz-Castillo, Dora Evelia Rodríguez-Félix, Heriberto Grijalva-Monteverde, Lauren Lucero Lizárraga-Laborín, María Mónica Castillo-Ortega, Teresa del Castillo-Castro, Francisco Rodríguez-Félix, Pedro Jesús Herrera-Franco, Materials. 8 (2015) 137-148.
Author Response
Open Review 2
(x) I would not like to sign my review report
( ) I would like to sign my review report
English language and style
( ) Extensive editing of English language and style required
( ) Moderate English changes required
(x) English language and style are fine/minor spell check required
( ) I don't feel qualified to judge about the English language and style
Yes | Can be improved | Must be improved | Not applicable | |
Does the introduction provide sufficient background and include all relevant references? | ( ) | ( ) | (x) | ( ) |
Is the research design appropriate? | (x) | ( ) | ( ) | ( ) |
Are the methods adequately described? | (x) | ( ) | ( ) | ( ) |
Are the results clearly presented? | ( ) | (x) | ( ) | ( ) |
Are the conclusions supported by the results? | (x) | ( ) | ( ) | ( ) |
Comments and Suggestions for Authors
The manuscript reported by the present authors investigated the preparation and characterization of PLA/HDPE in the presence of different coupling agents. The manuscript is well-done and arranged. The authors have done enough experiments to show the study is relevant and adequate for publications. Some points and clarifications, however, should be added and discussed to make this manuscript suitable for publication in Materials.
1. The mechanical results of PLA/PE blends with/without coupling agents should be discussed and compared with previous investigations in literature. The lack of discussion makes difficult to determine the scientific contribution of the present work.
The addition of Lotader AX8840 seems to be very effective to improve compatibility between HDPE and PLA, compared to other compatibilizing agents [26-27]. Significant improvement in elongation at break was displayed for all compatibilizer percentages, although the presence of a high percentage of brittle polymer like PLA.
2. It would be useful for the readers if the authors give more insights and explanations on the compatibilization mechanism.
Chemical blending has been added in the introduction “Usually, chemical compatibilization, obtained with the use of grafted polymers or random copolymers, is performed to reduce the size of dispersed phase thanks to functional groups reactivity. In this way, a reduction of interfacial tension and the coalescence impediment of the dispersed phase is possible [22].”
SEM “Reactive groups of compatibilizers can interact with hydroxyl and/or carboxyl groups of PLA, while ethylene chain of compatibilizers can easily be mixed with HDPE. These results are in agreement with FTIR analyses.” Explanation of possible mechanisms responsible for higher compatibilization
3. Relevant literature should be cited. The following examples are highly recommended;
[1] E. Javadi, A. Babaei, M. Nouri, Correlation of the morphological and mechanical properties of a biodegradable blend based on polylactic acid, Journal of Macromolecular Science, Part B: Physics. 56 (2017) 194-201.
[2] K. Hamad, M. Kaseem, M. Ayyoob, J. Joo, F. Deri, Polylactic acid blends: The future of green, light and tough, Progress in polymer Science. 85 (2018) 83-127.
[3] Jesús Manuel Quiroz-Castillo, Dora Evelia Rodríguez-Félix, Heriberto Grijalva-Monteverde, Lauren Lucero Lizárraga-Laborín, María Mónica Castillo-Ortega, Teresa del Castillo-Castro, Francisco Rodríguez-Félix, Pedro Jesús Herrera-Franco, Materials. 8 (2015) 137-148.
Suggested references have been added (15-16-17)